# A Method for Stress Detection Using Empatica E4 Bracelet and Machine-Learning Techniques

**DOI:** 10.3390/s23073565

**Published:** 2023-03-29

**Authors:** Sara Campanella, Ayham Altaleb, Alberto Belli, Paola Pierleoni, Lorenzo Palma

**Affiliations:** Department of Information Engineering (DII), Università Politecnica delle Marche, 60131 Ancona, Italy

**Keywords:** objective stress measurement, wearable sensors, machine learning, IoT, chi-square test, Empatica E4

## Abstract

In response to challenging circumstances, the human body can experience marked levels of anxiety and distress. To prevent stress-related complications, timely identification of stress symptoms is crucial, necessitating the need for continuous stress monitoring. Wearable devices offer a means of real-time and ongoing data collection, facilitating personalized stress monitoring. Based on our protocol for data pre-processing, this study proposes to analyze signals obtained from the Empatica E4 bracelet using machine-learning algorithms (Random Forest, SVM, and Logistic Regression) to determine the efficacy of the abovementioned techniques in differentiating between stressful and non-stressful situations. Photoplethysmographic and electrodermal activity signals were collected from 29 subjects to extract 27 features which were then fed into three different machine-learning algorithms for binary classification. Using MATLAB after applying the chi-square test and Pearson’s correlation coefficient on WEKA for features’ importance ranking, the results demonstrated that the Random Forest model has the highest stability (accuracy of 76.5%) using all the features. Moreover, the Random Forest applying the chi-test for feature selection reached consistent results in terms of stress evaluation based on precision, recall, and F1-measure (71%, 60%, 65%, respectively).

## 1. Introduction

One of the main factors contributing to both physical and mental illnesses in people is stress [1]. An organism’s natural reaction to an intrinsic or extrinsic situation, whether it be favourable or unfavourable, physical or mental, is known as stress [2]. It is the body’s method of coping with an oppressive or negative situation and constantly works to restore the body to its normal balance [3]. Stress-related pathologies or disorders are thought to be the second most common cause of disease in both Europe and the United States, accounting for three out of every four doctor visits [4].

The first stage of stress is the disruption of an organism by a stimulus or event known as stressors [3].

Although stressors can take on many different forms, they can be broadly divided into two categories: psychological and physiological. Psychological stressors include things such as debt, the death of a loved one, losing a job, studying for an exam, and other similar items. Physiological stressors include things such as infections, high temperatures, and a lack of relaxation. When the body perceives a situation as stressful, it can trigger short-term or long-term reactions. The hypothalamus in the brain plays a crucial role in this process by activating and sending signals to the pituitary gland, which then stimulates the adrenal gland to produce cortisol. This hormone helps to stabilize the blood sugar supply and restore the body to normal function. In addition, the adrenal medulla, which is part of the autonomic nervous system, is stimulated by the hypothalamus to produce short-term stress responses. This results in the release of adrenaline, which causes the fight-or-flight response and activates the sympathetic nervous system. Once the stressor is removed and the parasympathetic nervous system takes over, the body returns to its normal state [5].

Based on the time-lapse, stress can be divided into three categories and each of them has a unique set of symptoms, traits, duration, and treatment options. It is distinguished into acute stress, the most common, characterized by short duration and associated with negative thoughts, episodic stress, which happens when intense stress is sustained over a long period before it becomes a habit, and chronic stress, which might be the result of early childhood experiences and traumatic experiences from the past that have shaped one’s life [6].

Psychometric tools, scales, questionnaires, or surveys were used as part of the conventional method of stress detection. Although they are inexpensive and simple to use, questionnaires have some drawbacks that make them less useful since they are based on individual perceptions [7]. Studies have revealed that, in addition to the conventional methods of detecting stress through questionnaires and behavioral observations, it can also be determined and measured from physiological, psychological, and neurological responses [8]. Heart rate variability (HRV), galvanic skin response (GSR), respiratory rate, blood oxygen saturation, cortisol level, blood pressure (BP), and brain signals are indicative parameters because they are connected to the autonomic nervous system [9]. Smart wearable devices that can measure signals even in natural settings for assessing cognitive and sensory states have been made possible by recent advancements in embedded systems and sensors. Presently, these vital signals are collected using several variegate wearable devices—smart watches, chest belts, smart t-shirts, and head-mounted devices [10]—allowing ongoing mental health monitoring to be easier compared to the past. The widespread market adoption of smart wearables has given people the ability to track, store, and transfer personal information about their surroundings, physical activity, and health [11].

Stress is a heterogeneous disease that affects adults and young people equally. Due to the demanding physical and mental efforts required of employees, the workplace has become a major source of stress in the latest days [12]. It could also be a result of staff not having the resources they require to do their jobs well or of staff not having their needs met. Stress at work has been linked to frequent absences, mistakes, and lower productivity [13]. According to evidence, the EU spends about EUR 617 billion a year on social welfare, health care, and programs to help people who are stressed out or depressed at work [14]. This demonstrates how stress at work not only affects the productivity of individuals but also the entire state. Teenagers frequently experience academic stress, a type of mental distress brought on by the many expectations that are placed on them. It can be difficult to avoid stress as a factor. Students experience stress due to a variety of demands, including homework, exams, classes, projects, friends, and family. Their academic success is directly correlated with these demands. Students under high stress often experience depression and anxiety [15].

Using the physiological data that was recorded during stressful situations, the proposed work seeks to automatically identify a person’s state of stress. Such detection can aid in the monitoring of stress and the prevention of harmful diseases linked to stress. This is even more important when considering workers. Monitoring employee stress levels is a crucial component of fostering a healthy and effective work environment. Therefore, this study aims to analyze signals collected by the Empatica E4 bracelet, using machine-learning algorithms, based on our protocol for data pre-processing and to individuate the accuracy of the techniques mentioned above in distinguishing between stress and no-stress situation. Our protocol was created to simulate a demanding situation that might arise in a workplace where several tasks must be completed simultaneously. Additionally, we want to see if the quality of the bracelet’s recorded data is sufficient for a classification problem and can therefore be used in a real scenario despite the various arm movements and abrupt changes between activities that can affect significantly the overall monitoring. The following two subsections describe the physiological signals collected and their meaning while the other one reports related works in the same fields. In Section 2 and Section 3, the procedure used to collect and analyze data and the obtained results are reported, respectively. Finally, in Section 4 and Section 5 a detailed explanation of the results and the pros and cons of this study are shown.

### 1.1. Physiological Signals for Stress Assessment

Strong evidence from research suggests that physiological signals carry information about human emotions [16]. A fast heartbeat, excessive sweating, and unusual facial expressions are typical manifestations of emotion, which is an intense mental experience [17]. According to the results shown in [18], a lower HRV is linked to the feeling of “happiness”, whereas a higher value is linked to “joy” or “amusement.” In a similar vein, research has also been done on the function of GSR. It is shown that GSR can distinguish between the emotions of “fear” and “anger”, “fear” and “sadness”, and “happy” and “sad” [19]. As previously said, several physiological signals can be used, but for the aim of this study, we will consider only electrodermal activity (EDA) and photoplethysmogram records (PPG). Specifically:PPG: measures the blood volume pulse (BVP) which changes for each interbeat interval due to alterations in blood volume and blood pressure in response to stress stimuli. PPG works by emitting light from a source and measuring how much of it is absorbed by the blood based on the amount of blood in the volume. PPG devices are commonly used to calculate heart rate (HR), heart rate variability (HRV), and related information.EDA: The sympathetic innervation of sweat glands results in electrodermal activity (EDA), which is a measure of changes in the skin’s electrical conductance. EDA is detected through the modulation of the conductance of an applied current by sweat gland activity. Increased sweating enhances the electrical conductivity of the skin due to the presence of water and other substances in sweat, including minerals, lactic acid, and urea. EDA is more sensitive to psychological stimuli compared to thermal stimuli, making it a potential measure of sudomotor activity and an unbiased evaluation of arousal [20].

### 1.2. Related Works

Mental stress has long been recognized to have harmful effects on human health. Continuous mental stress, for instance, can lead to physical and mental conditions such as cardiovascular diseases, hypertension, diabetes, cancer, headaches, depression, anxiety, and insomnia. Early detection of high stress levels is necessary to stop these harmful effects. Several recent stress-detection strategies have been put forth, typically based on machine-learning (ML) methods. The work of Kyriakou et al. [21] in 2019 aimed to bridge the gap between laboratory settings and real-world field studies by introducing a new algorithm to detect moments of stress (MOS) using wearable physiological sensors. Eleven subjects wore an Empatica E4 device and were subjected to a laboratory experiment, an auditory stimulus was used to induce stress. Furthermore, to validate the algorithm, a real-world urban experiment was introduced. An accuracy of 84% was obtained using the proposed algorithm. The study carried out by Kaczor et al. [22] aimed at the objective measurement of physician stress in the emergency department using the Empatica E4 smartwatch. EDA, acceleration, and heart rate signals were acquired from eight participants during clinical shifts (typically 8–10 h). After that several machine-learning classifiers were used and the best accuracy obtained was 70% to detect stress during the working shift with respect to the baseline condition. In the study proposed by Dai et al. [23], 32 subjects participated in 2 h of laboratory and 24 h of field-based experiments. The aim of the study was comparing between objective and subjective stress-detection models. In particular, participants were given a mental mathematics assignment, which required them to solve a series of complex mathematical problems over a given period of time. Support Vector Machine, Random Forest, AdaBoost, Gradient Boosting, and Logistic Regression classifiers were used to detect stressed or non-stressed periods in both objective and subjective stress models. In the study carried out by Mach et al. [24] in 2022, a laboratory experiment consisting of an arithmetic task which is counting down or up steadily, and physical activity (sitting vs. stepping) with 52 participants was conducted. This study aimed to assess mental workload via heart rate measurement and a chest strap with a 1-channel ECG. They found that the mean heart rate increased when participants performed the arithmetic task compared to the conditions with no arithmetic task while sitting and stepping. In the study done by Seo et al. [25] in 2022, 24 participants wore a Zephyr chest strap equipped with a BioHarness module to acquire ECG and Respiratory signals. Furthermore, the subjects were sitting in front of a laptop and faced a camcorder screen to register facial information. The experiment lasted for 45 min and comprised two stages: an initial setting stage, and an actual experiment stage which is the Stroop task. The actual experiment consists of Relax, Easy Stroop, Recovery, Hard Stroop, and Recovery, 5 min for each. Afterwards, signal and image processing was done followed by a Deep Neural Network (DNN) classifier. The accuracy for two or three levels of stress classification was 73.3%, and 54.4%, respectively.

In a study by Chalabianloo et al. [26] in 2022, 32 subjects were subjected to a laboratory experiment that consisted of baseline, stress, recovery, and cycling sessions. Stress sessions were performed using the Stroop task and different physiological signals were recorded using seven different wearable devices simultaneously. The best accuracies across most of the devices were obtained using an Extremely Randomized Tree classifier, for example, 88.26% for the BITalino device. Furthermore, to study the effects of multimodality, the EDA signal was introduced using Empatica E4. After that, the same classifiers mentioned above were used. The accuracy obtained considering only HR was 83.89% using the Random Forest classifier, while when considering HR and EDA the accuracy became 90% using the Extremely Randomized Tree classifier.

In the study done by Suni Lopez et al. [27] a laboratory experiment was conducted to detect stress in the office workplace, the experiment consisted of interacting with a laptop where the Stroop task was installed. Twelve subjects participated and were asked to wear the E4 smartwatch to collect EDA data, and headphones to interact with the environmental trigger (fire alarm). After signal filtering, aggregation, and discretization, an accuracy of 79.17% was obtained using statistical method classification.

In Table 1, all the mentioned works together with the devices and protocol applied are summarized.

## 2. Materials and Methods

This work’s primary objective is to analyze data collected by Empatica E4 and to assess the validity of our model, using machine-learning techniques.

The data have been collected in our laboratory after the candidates have been instructed about the protocol, the aim of this study, and signed privacy questionnaires. A total of 29 subjects were individuated and equipped with Empatica E4. The chosen environment for this study is Matlab 2022.

This section is divided into five subsections, namely Empatica E4, Data Acquisition Protocol, Data Pre-Processing, Features extraction, and machine-learning algorithms, each dedicated to a specific portion of the work carried out.

### 2.1. Empatica E4

The study used the Empatica E4 bracelet, a wearable device that collects continuous and instantaneous physiological data through its four sensors: temperature sensor, accelerometer, EDA sensors, and PPG sensors. The E4 was worn snugly on the wrist to ensure stability during testing. Data were stored through Bluetooth streaming acquisition and monitored in real time through the E4 Realtime app on a smartphone. The data were then uploaded to Empatica’s cloud platform, E4 Connect, for pre-processing.

### 2.2. Data Acquisition Protocol

In order to evaluate mental stress, a protocol must be defined and appropriate stressors should be identified. Several categories can be individuated, such as cognitive stressors, characterized by tasks that require a high level of attention, concentration, and memory, such as solving complex mathematical problems or memorizing long word lists. Social stressors can be perceived as threatening or judgmental, such as participating in a job interview or giving a public speech. Physical stressors are those situations that require intense physical effort, such as engaging in high-intensity exercise or being exposed to extreme temperatures. Finally, we can distinguish between emotional stressors, which can elicit intense and negative emotions and psychological stressors, which require experiencing a sense of uncertainty or lack of control [28].

On the basis of the different stressors, we searched the literature to recreate a protocol that combined all these stressors to create more complex and realistic mental stress. We focused on creating mainly cognitive, social, and physiological stressors since they are the most likely to be triggered in a working environment and easy to induce cognitive load in a laboratory situation.

Therefore, we came out with the protocol depicted in Figure 1, developed based on the one suggested in [29]. We decided to apply this protocol as a base to develop our own due to the high accuracy that the just quoted study reached.

Three minutes of rest were recorded after the bracelet was turned on to establish a baseline. After each task, a 2-minute rest period was carried out. In the first task, participants had ten minutes to construct a Lego object using only the images printed on the box and no instructions. The second task is to assemble the same Lego creation within five minutes, but this time with the aid of the instructions. The third task requires the participant to assemble another Lego creation made of larger pieces in three minutes while following instructions and counting backwards from 180 (the total amount of time available to complete the task) to zero. Each of the aforementioned tasks was developed to simulate manufacturing activities such as assembly and manual handling and to induce the mental stress that workers can face while doing a specific job. The fourth test is entirely mathematical and involves repeatedly subtracting backwards the number 13 from 511. There is no time limit in this situation. This task is inspired by the Montreal Imaging Stress Task, created to investigate the effects of psycho-social stress in the human brain [30]. The fifth and final test requires the subject to give a one-minute oral presentation of themselves and their resume, since it has been demonstrated that an oral presentation can cause stress and memory impairments [31].

### 2.3. Data Pre-Processing

After the data acquisition, we clean our data to extract features and apply any machine-learning algorithm.

The duration of the signal segment is known to affect HRV and pulse rate variability (PRV) features [32]. This means that the features will differ depending on the length of the segment under consideration. This implies that the HRV features are contingent upon the length of the segment under consideration, In this study, a pre-processing methodology was employed that initiated the segmentation of the PPG and EDA signals into intervals of 1-minute duration. Several factors influenced the decision to use a 1-minute interval. To begin, the data collection protocol for this study included a 1-minute task—the CV presentation task. Second, a one-minute duration is appropriate for use in wearable health monitoring devices. Third, to maximize data segments. The segmentation process included all 29 subjects who took part in this study. For each BVP and EDA signal, a total of 1068 data segments were extracted. Following that, the segments were labeled according to the tasks or rest period.

Regarding the PPG signals, different noises and artifacts can affect the signals during PPG recording, lowering the stress-detection system’s accuracy. The most prevalent of them is the motion artifact, which has a significant impact on the PPG signal quality. For this reason, all the segments were filtered using a Chebyshev II order-4 filter with a stopband attenuation of 20 dB and a passband of 0.5–5 Hz [33]. The crucial step is pinpointing the peak of the PPG signal and the distance between two consecutive peaks. Therefore, peak detection was performed using the *findpeaks* function, with a threshold set to a minimum peak distance of 0.4 s and a minimum peak height of 0. Afterwards, peak-to-peak matrices were calculated by subtracting every two consecutive peaks. Following these computations, only intervals with a time duration of 500 to 1200 ms (corresponding to heart rates of 120 and 50 beats per minute) were taken into consideration, while all abnormal intervals (time duration less than 500 ms or greater than 1200 ms) were excluded. These limits were chosen based on the Work of Zubair et al. [34] but we modified the lower limit to 500 ms because it produces 120 Bpm instead of 600 ms which corresponds to 100 Bpm. This means that if we choose 600 ms, all HR more than 100 Bpm will be eliminated, removing HR values associated with stress tasks. However, excluding too many abnormal intervals would reduce the length of the PRV series. PPG segments with abnormal intervals that made up less than 15% of all intervals were therefore taken into account. The threshold of 15% was selected to ensure that the selected PPG segment still has a time length greater than the 50 s after removing abnormal intervals [34]. As a result, the total number of PPG segments was reduced to 843, obtaining 320 segments for the rest condition and 523 for all the tasks.

For what concerns EDA pre-processing, upsampling from 4 to 64 Hz was performed to make both signals at the equal sampling frequency [35]. To remove any artifacts, smoothing using the Gaussian low pass filter, with a 40-point window and sigma of 400 ms, was carried out [36,37]. Finally, all the clean segments went through the feature extraction process. In Figure 2, there is a schematic representation of the PPG and EDA signal processing, while in Figure 3 are visible the signals before and after the cleaning.

### 2.4. Features Extraction and Selection

Meaningful information was extracted from each data segment during the features extraction phase to characterize the various data portions in the time and frequency domains. Table 2 lists the 27 features that were chosen to quantify our data after being successfully applied in earlier studies for both PPG and EDA signals. For the BVP, a total number of 16 features were extracted, in particular, the features could be divided into two categories: first one PRV based on calculated peak-to-peak (PP) matrices and it is worth mentioning that only consistent features in ultra-short-term matrices were included [32]. The second one is related to the signal itself such as mean, median, mode, minimum, maximum, standard deviation, mean, and standard deviation of the first and the second derivative of the filtered signal [35,36]. To extract this information, an algorithm was developed using several functions available on Matlab Statistics and Machine-Learning Toolbox. Below are reported the mathematical formula for the statistic features and the ones computed in the frequency domain. Equation (Equation 1) shows the formula for the mean computation, while Equations (Equation 2) and (Equation 3) represents the median and standard deviation, respectively. In Equation (Equation 4) the absolute power in high frequency is reported, where f(λ) is the power spectrum of the PP tachogram [38]. Finally, Equation (Equation 5) is based on the summation of successive PP intervals such as a moving average. Its deviation represents the “long-term HRV” [38].
(1)$$\bar{x}=\frac{1}{n}\sum_{i=1}^{n}x_{i}$$
(2)$$\left\{\begin{array}{cc} x_{(k+1)/2} & n\;\mathrm{odd}\\ \frac{x_{n/2}+x_{n/2+1}}{2} & n\;\mathrm{even} \end{array}\right.$$
(3)$$SD=\sqrt{\frac{\sum_{i=1}^{n}{\left(x_{i}-\bar{x}\right)}^{2}}{n-1}}$$
(4)$$HF=\int_{0.15\mathrm{Hz}}^{0.40\mathrm{Hz}}f\left(\lambda \right)\;d\lambda$$
(5)$$SD2=\sqrt{\frac{1}{2}}\cdot \mathrm{std}\left(PP_{i+1}+PP_{i}\right)$$

For the maximum and minimum values of each signal, the functions *min[x]* and *max[x]*, with *x* as a signal, from the previously mentioned toolbox, were applied.

The BIO-SP tool was used to extract skin conductance response (SCR) features. SCRs are commonly found in electrodermal activity signals and can be identified using differentiation and convolution with a 20-point Bartlett window. This method is commonly used in EDA signal analysis to identify and characterize SCRs, which are important indicators of sympathetic nervous system activity. All the features available in this tool were extracted, including the mean rise time, duration, amplitude, number of peaks, and mean of the SCR signal [37]. In the end, a feature standardization using the Z-score was performed since the parameter magnitudes were different.

A key idea in modeling is feature selection, which can improve a model’s performance by eliminating unnecessary features. Feature selection becomes one of the crucial steps in building our stress-detection model to reduce the complexity and the time needed for the execution of computations, which have been greatly increased due to the use of cross-validation. To enhance the effectiveness of stress detection, the most pertinent and significant features should be chosen. The ranking of feature importance was performed using two methods: Univariate feature ranking for classification using chi-square tests (chi-test), in Matlab, and the Pearson’s correlation coefficient with the Waikato Environment for Knowledge Analysis (WEKA) [39]. The former one is so-called because it is conducted on two distributions two determine the level of similarity of their respective variances. In its null hypothesis, it assumes that the given distributions are independent [40]. The Chi-square test can be written as
(6)$$\chi^{2}=\sum \frac{{\left(O-E\right)}^{2}}{E}$$
where $\chi^{2}$ represents the calculated value of the chi-square test, ∑ denotes the sum, *O* represents the observed number of events in each category, *E* represents the expected number of events in each category, and ${\left(O-E\right)}^{2}$ represents the squared difference between the observed and expected number of events in each category. In our case, using the Matlab function *fscchi2*, which examines whether each predictor variable is independent of a response variable using individual chi-square tests. A small *p*-value of the test statistic indicates that the corresponding predictor variable is dependent on the response variable, and, therefore, is an important feature. We computed the predictor scores as –log(*p*), with *p* being the *p*-value. Therefore, a large score value indicates that the corresponding predictor is important. Then, we computed the mean value of the score and used it as a threshold.

Then, using WEKA, we applied a Pearson correlation coefficient to create rankings for each feature. The Pearson’s correlation coefficient is a measure of the linear relationship between two variables, X and Y. It ranges between −1 and +1, where −1 indicates a perfect negative linear relationship, 0 indicates no linear relationship, and +1 indicates a perfect positive linear relationship. The formula for the Pearson correlation coefficient is
(7)$$r_{xy}=\frac{\sum_{i=1}^{n}\left(x_{i}-\bar{x}\right)\left(y_{i}-\bar{y}\right)}{\sqrt{\sum_{i=1}^{n}{\left(x_{i}-\bar{x}\right)}^{2}\sum_{i=1}^{n}{\left(y_{i}-\bar{y}\right)}^{2}}}$$
where $r_{xy}$ represents the Pearson correlation coefficient between X and Y, ∑ denotes the sum, *n* is the sample size, $x_{i}$ and $y_{i}$ are the ith observations of X and Y, respectively, $\bar{x}$ and $\bar{y}$ are the sample means of X and Y, respectively. In our case, the function CorrelationAttributeEval was applied to evaluate the worth of an attribute by measuring the correlation between it and the class. Any attributes with rankings below a cutoff of 0.10 were eliminated [36].

### 2.5. Classification

A class label related to the presence or absence of stress is returned from the ML classifiers using the subset of features produced, as well as the total set of features, as input. Based on the literature review reported in Section 1.2, the most frequently used and effective binary classifiers for identifying stress have been implemented. In particular, Random Forest and Logistic Regression, and SVM with cubic kernel on MATLAB.

As its name suggests, the Random Forest is made up of numerous individual decision trees that work together as an ensemble. Every single tree in the Random Forest spits out a class prediction, and the classification that receives the most votes becomes the prediction made by our model. Logistic Regression is a statistical approach that is used for classification problems and is based on the concept of probability. It is used when the dependent variable (target) is categorical. Finding a hyperplane in an N-dimensional space (N is the number of features) that categorizes the data points is the goal of the SVM algorithm. These three approaches were tested on Classification Learner on Matlab. The hyperparameters were tuned for SVM and Random Forest. To obtain the best metrics, we vary the number of splits and learners during various trials for Random Forest. Although the learners range from 30 to 100, the number of splits varies from 500 to 1000. The number of splits allows us to specify the maximum number of branch points to control the depth of our tree while the number of learners is the number of decision trees in the RF: the greater the value, the greater will be the number of subsets the data will be divided into to train every decision tree. To achieve the best results for the SVM model, the cubic kernel was used and several values of the kernel scale from 1 to 10 were applied.

The 10-fold cross-validation configuration setting was used to test the machine-learning algorithms for the model evaluation. In this configuration, the new features dataset was divided into 10 subsamples randomly, with 9 subsamples serving as training data and 1 subsample serving as validation data. The resulting accuracy percentage is the average over the 10 iterations using the available subsamples as validation data. The ability to categorize the presence or absence of stress, as a binary classification task, was assessed using the classification performance metrics of accuracy (Equation 8), precision (Equation 9), recall (Equation 10), and F-measure (Equation 11):(8)$$Accuracy=\frac{TP+TN}{TP+TN+FP+FN}$$
(9)$$Prec=\frac{TP}{TP+FP}$$
(10)$$Rec=\frac{TP}{TP+FN}$$
(11)$$F1=\frac{2*Prec*Rec}{Prec+Rec}=\frac{2*TP}{2*TP+FP+FN}$$

## 3. Results

Empatica E4 data were pre-processed and analyzed to extract features from each recording. The features chosen for feeding the ML algorithms are then reported for both methods, in Figure 4. Applying the chi-test method, only 10 features were chosen from the original 27 ones while using the Pearson correlation coefficient only 15 features fed the ML algorithms. Through both methods, it can be seen that all HRV features exhibited values above the selected threshold in Pearson’s correlation method. Additionally, in the chi-square method, four of the HRV features surpassed the threshold confirming the validity and stability of the information that they carry. Using different ML algorithms, this feasibly ideal combination of features, composed of the parameters more responsive to stress, was tested for correctly discriminating the absence or existence of stress. Moreover, to validate the feature extraction process, we also fed the algorithms with all 27 features.

For said binary classification, three distinct machine-learning classifiers had been used, each trained with a 10-fold cross-validation strategy. Thus, every algorithm was evaluated in terms of accuracy percentage, Sensitivity, precision, and F1. The results are reported in Table 3. These metrics can be influenced by several factors, such as the size and quality of the dataset, the features used for the model, the type of stressors, and the specific algorithm used. Comparing the outcomes of the full set of features and the selected features, it is apparent that the accuracy, as well as other performance metrics, decrease for the Logistic Regression. This decline may be attributed to the unavailability of hyperparameter tuning within the MATLAB toolbox. Conversely, the SVM and RF exhibit consistent outcomes even after feature reduction, possibly due to the flexibility of modifying hyperparameters in different situations. In the case of the full set of features, the SVM was fitted with a cubic kernel, box constraints equal to 1, and a kernel scale of 7. Meanwhile, the RF was constructed using 1000 splits and 50 learners. However, for the selected subsets, the parameter settings were different. In particular, when using the Chi-test and Pearson’s correlation coefficient features to feed the SVM, the same cubic kernel and box constraints were used, but the kernel scale was adjusted to 3. For the RF, the best performance was obtained with 500 splits and 100 learners when using the chi-test protocol, and with 1000 splits and 50 learners when using the Pearson correlation coefficient.

The Random Forest approach is more stable and reliable while working with an imbalanced dataset, discriminating correctly between stress and no-stress situations. The Precision and recall are higher in both cases, especially with the chi-test approach, being consistent also for the label 0, where the other classifiers demonstrate to be more affected by the unbalancing. However, Logistic Regression and SVM also achieve good results in some metrics, indicating that they could be suitable for this type of dataset. These outcomes are also visible in the confusion matrices in Figure 5.

## 4. Discussion

A new system was proposed to analyze physiological signals measured with a wearable device on a test population before and after performing tasks designed to induce mental stress.

The Pearson coefficient was used for feature selection, and the results showed that most of the features were related to the PPG signal, while only two were related to the SCR signal. The top three features were the standard deviation of HR, PP, and long-term variability (SD2), respectively, with a ranking of over 0.25, indicating the importance of HRV analysis in detecting stress, consistent with previous research. For the BVP signal, the standard deviation of the first derivative, the standard deviation of the signal itself, and the standard deviation of the second derivative were the next three important features, respectively, with a ranking of over 0.1, suggesting that dispersion is more important than average values. As for EDA, only the mean of SCR and the mean amplitude of SCR, which are related to the phasic component of the EDA signal, had a rank higher than the threshold.

In the chi-square approach, among the six HRV features, four were found to be above the predetermined threshold. Moreover, the standard deviation of PP and the standard deviation of HR were identified as the two most significant features. Regarding the BVP signal, similar to the Pearson correlation approach, the standard deviation of the second derivative, first derivative, and the signal itself were identified as important features. However, their mean values were not found to be significant. In the case of EDA signal analysis, it was found that only the number of SCR peaks was important with respect to the threshold. This observation confirms that the number of peaks (N_Peaks) is the primary indicator of sympathetic nervous system (SNS) activity, and, thus, related to a stress condition [41].

Focusing on the classification aspect, in general, our analysis indicates that the classifiers’ accuracy consistently exceeds 70%. This suggests that the pre-processing and original feature selection were appropriate for the database under consideration.

Tuning the hyperparameters had a positive impact on the results. Notably, the Logistic Regression approach lacked the capacity for optimization in MATLAB, which was highlighted when transitioning from the full set of features to the reduced set. In comparison to RF and SVM, where reducing the number of features had a minor effect on the results. The accuracy, along with other metrics of LR, was less promising, particularly for label 0, where precision and recall values were 0.68 and 0.53, respectively. On the other hand, feature reduction not only reduced computational costs but also stabilized and improved the results.

Overall, the Random Forest algorithm consistently exhibited superior performance compared to other classification methods. This could potentially be attributed to the fact that Random Forest classifiers rely on randomness, which promotes more generalized modeling. This observation is corroborated by the precision, recall, and F1-measure metrics, which demonstrate the algorithm’s effectiveness when contrasted with SVM and LR matrices.

The findings of the current study are consistent with the existing literature in the field. Specifically, the results of the feature selection are aligned with the other studies that have emphasized the importance of the selected features. It is noteworthy that despite the differences in the devices employed in the previous studies, the HRV-based features have emerged as the most robust indicator of stress, along with the SCR information [24,36]. The results obtained from the current study indicate that both feature evaluation methods employed, particularly the chi-test method, possess considerable strength in selecting stress-related characteristics. The outcomes achieved with the chi-test method align with the ones obtained by [42]. Even if they applied different classifiers, the results are consistent with ours, demonstrating that the chi-test method is feasible for mental stress detection. As for the machine-learning (ML) outcomes, the Random Forest classifier surpasses other techniques when applied to data acquired using Empatica E4. According to a study proposed by [43] the RF is a dependable classifier for binary classification, demonstrating high accuracy scores and outperforming the SVM one, which is consistent with our findings. Furthermore, when comparing our results to studies that employ data collected in real-world settings, the RF method achieves notable accuracy scores, despite experiencing lower precision values [44]. This implies that RF has a high accuracy for detecting stress-related data. However, it was found that RF misclassified a considerable number of non-stress-related data as stress. In a study conducted by Cosoli et al. [36], suggested that the SVM and LR algorithms are more reliable in classifying and detecting stress events compared to our results, where both approaches yielded lower levels of accuracy. It is plausible that such differences in the findings may be attributed to variations in the stimuli employed to induce stress, as well as the dissimilar pre-processing steps applied to clean the data.

Observing the performance of the classifiers in detail, it becomes apparent that the classification of the presence of stress (label 1) outperforms the classification of its absence (label 0). This discrepancy may be primarily attributed to the unequal number of segments, as the number of segments associated with stress is greater than those associated with the rest phase (523 vs. 320, respectively). This is because, in the overall protocol, the duration of the tasks is greater than the total rest period. Additionally, the time allocated for rest periods may not be sufficient for the participants to achieve a completely stress-free state during the inter-task rest periods, further complicating the classification task. It is worth noting that the remaining classifiers exhibited lower values in all of the evaluated metrics compared to the Random Forest, particularly the LR classifier, where no hyperparameters have been set for this classifier, although the results are still acceptable.

Despite the fact that the segments of rest and stress conditions were unbalanced, our results were still able to distinguish between these two conditions with a reasonable degree of accuracy. However, our findings fell short of those reported in previous literature, and there is a possibility for improvement for example by balancing the data using different algorithms as the study carried by [45] suggests. They obtained higher performance and accuracy after manipulating the data with ADASYN.

One feasible explanation for the suboptimal performance of our system is the presence of noise associated with the Empatica E4 device, particularly during hand movement tasks. Another factor to consider is the relatively small sample size used in our study. Increasing the size and diversity of the participants would help to enhance the generalizability of our findings and improve the accuracy of our models. A small dataset may not be representative of the broader population and may be prone to inaccuracies and erroneous conclusions, as it could be influenced by outliers or anomalous data. Furthermore, the decision to conduct our study in a laboratory setting may have limited our ability to simulate real-world working conditions. In the future, additional stimuli could be introduced to overcome this limitation and more accurately replicate real-world scenarios.

Despite the drawbacks described above, our approach achieved high performance in the detection the stress situations. This means that our choices for data manipulation and feature selection are sufficiently strong to deal with an unbalanced dataset. This means that in a real situation where motion artifacts have higher intensity and unpredictable stressful situations can arise, stress can be detected and monitored to avoid any psycho-physical complications.

## 5. Conclusions

The object of this study was to assess the stress level by the measure of different physiological signals using a wearable sensor, Empatica E4. To address the stress condition, we have defined a new acquisition protocol based on the literature reviews. We hired 29 subjects and the acquisitions were performed in a laboratory environment. Moreover, the machine-learning approaches were developed accordingly with the best-performing algorithms in this field as well as the features chosen to characterize the model. Despite the limited baseline for the rest condition, which affected the balancing of the database, we reached an accuracy of 76.5%, 75.4%, and 75.7%, using all the features and both Pearson and chi-test approaches, respectively. To validate our model in the future, we suggest increasing the population size to include a diverse age range and implementing a new protocol that ensures a consistent baseline to avoid any misclassification issues. Additionally, incorporating multi-level stress tasks with different stressors, including real-life scenarios, could improve the model’s robustness. It would be valuable to compare our results with feedback from participants, obtained through the use of questionnaires. Moreover, we recommend considering various wearable devices available on the market to assess the impact of their characteristics on the results.

## Figures and Tables

**Figure 1 sensors-23-03565-f001:**
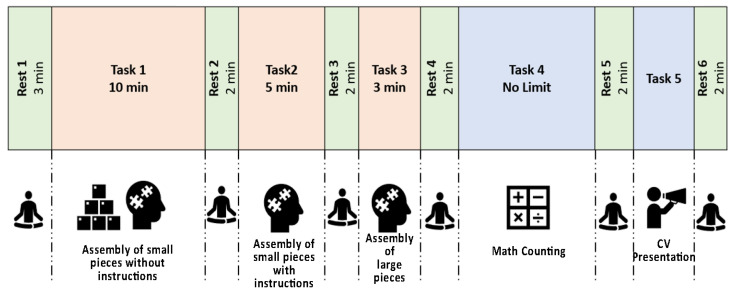
Data acquisition protocol carried out for each of the participants.

**Figure 2 sensors-23-03565-f002:**
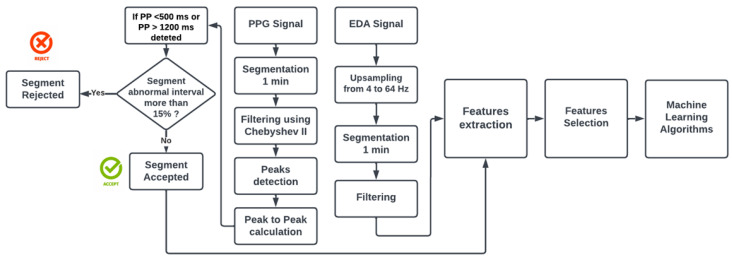
Flowchart for PPG and EDA pre-processing.

**Figure 3 sensors-23-03565-f003:**
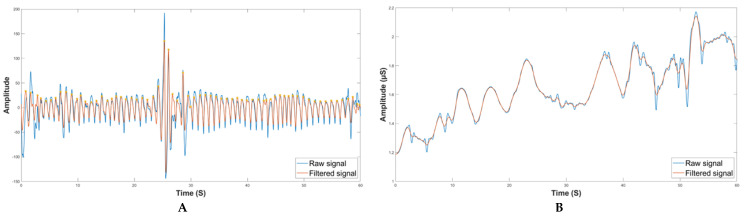
Data cleaning for both PPG and EDA records with the corresponding raw and clean signals. (**A**) Raw and clean PPG signal. (**B**) Raw and clean EDA signal.

**Figure 4 sensors-23-03565-f004:**
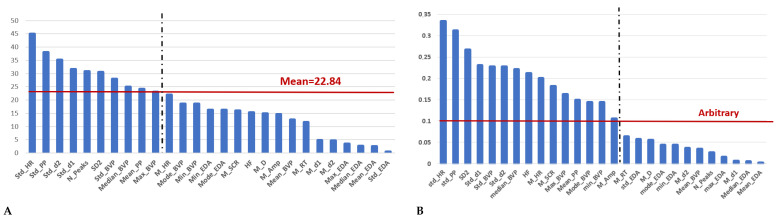
Ranks listed in order of importance for each feature extracted: (**A**) Chi-test method. (**B**) Pearson’s correlation coefficient.

**Figure 5 sensors-23-03565-f005:**
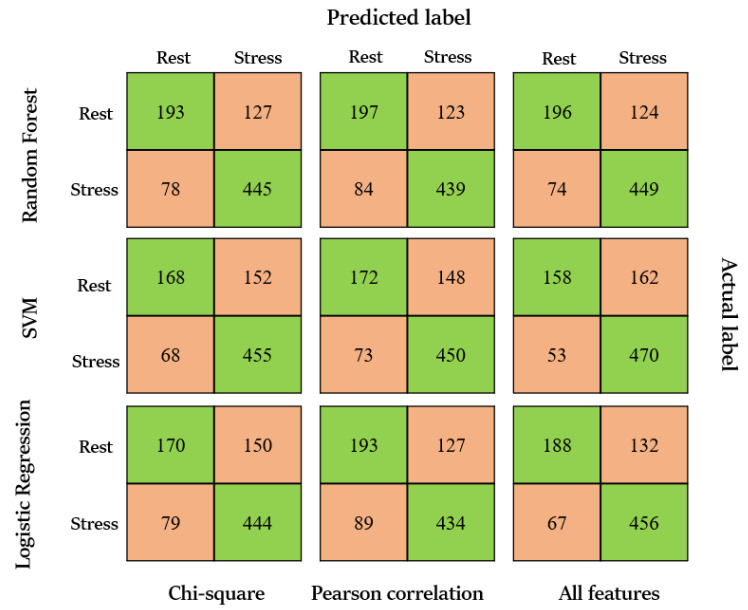
Confusion matrices for all the three machine-learning techniques and before and after the features’ selection.

**Table 1 sensors-23-03565-t001:** A list of the cited works with the protocol tasks used and the performances of the machine-learning algorithms. Abbreviation: HR: Heart Rate; ECG: Electrocardiogram; Resp: Respiration; ST: Skin Temperature; GSR: Galvanic Skin Response; ACC: Acceleration; PPG: Photoplethysmogram; DNN: Deep Neural Network; RF: Random Forest; DA: Discrimination Analysis; ERT: Extremely Randomized Tree; SVM: Support Vector Machine; K-NN: K-Nearest Neighbor; NB: Naïve Bayes; LR: Logistic Regression; DT: Decision Tree. * 7 devices are BITalino (r)evolution board, Firstbeat Bodyguard2, Polar H10, Zephyr HxM, Empatica E4, Samsung Gear S2, and CoreSense.

Study	Device	Signals	Method	Stressor	Accuracy
[21]	Empatica E4	GSR, ST	New Algorithm	Audible, real-world urban	84%
[22]	Empatica E4	GSR, ACC, HR	DT, LR, NB, SVM, K-NN, DA	Physicians in the emergency department	70%
[23]	Fossil Gen4	PPG, ACC	SVM, RF	Speech, Math, cold, daily life	82.6% (laboratory) 79.8% (daily life)
[24]	Samsung Gear S3	HR, ECG	Statistical	Math, Stepping	-
[25]	Zephyr	ECG, Resp	DNN	Stroop	73.3%
[26]	7 devices *	ECG, HR, GSR	ERT, RF	Stroop, cycling	83.89% (E4, HR) 90.62% (E4, HR & GSR)
[27]	Empatica E4	GSR	Statistical	Stroop, audible	79.17%

**Table 2 sensors-23-03565-t002:** All the features computed with their domain and abbreviation.

Signal	Domain	Features	Abbreviation
PPG	Time	Mean PPI, standard deviation of PP interval, mean heart rate, standard deviation of heart rate.	Mean_PP, std_PP, M_HR, std_HR
	Frequency	Absolute power in high frequency [0.15–0.4 Hz].	HF
	Non-Linear	Heart long-term variability.	SD2
	Statistical	Mean of the filtered signal, median of the filtered signal, mode of the filtered signal, minimum of the filtered signal, maximum of the filtered signal, the standard deviation of the filtered signal, mean of the first derivative of the filtered signal, the standard deviation of the first derivative of the filtered signal, mean of the second derivative of the filtered signal, the standard deviation of the second derivative of the filtered signal.	Mean_BVP, Median_BVP, Mode_BVP, Min_BVP, Max_BVP, Std_BVP, M_d1, Std_d1, M_d2, Std_d2,
EDA	Statistical	Mean, median, mode, maximum, minimum, standard deviation.	Mean_EDA, Median_EDA, Mode_EDA, Max_EDA, Min_EDA, Std_EDA
SCR	Time	Mean Duration, Mean Amplitude, Mean Raise Time, Number of peaks, Mean.	M_D, M_Amp, M_RT, N_PEAKS, M_SCR

**Table 3 sensors-23-03565-t003:** Performance metrics before and after applying the Chi-test and Pearson’s correlation coefficient methods for all the three machine learning techniques.

ML Algorithm	Chi-Test Method					Pearson’s Correlation Coefficient					All Features				
	Accuracy	Label	Prec	Rec	F1	Accuracy	Label	Prec	Rec	F1	Accuracy	Label	Prec	Rec	F1
Random Forest	75.7%	0	0.71	0.60	0.65	75.4%	0	0.70	0.62	0.66	76.5%	0	0.73	0.61	0.66
		1	0.78	0.85	0.81		1	0.78	0.84	0.81		1	0.78	0.86	0.82
SVM	73.9%	0	0.71	0.52	0.60	73.8%	0	0.70	0.54	0.61	74.5%	0	0.75	0.49	0.50
		1	0.75	0.87	0.81		1	0.75	0.86	0.80		1	0.74	0.90	0.81
Logistic Regression	72.8%	0	0.68	0.53	0.60	72.7%	0	0.68	0.53	0.59	76.4%	0	0.74	0.59	0.66
		1	0.75	0.84	0.79		1	0.75	0.85	0.80		1	0.78	0.87	0.82

## Data Availability

The data presented in this study are available from the corresponding author upon reasonable request.

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
