# Peer review of "A Method for Stress Detection Using Empatica E4 Bracelet and Machine-Learning Techniques"

_sensors, 2023, doi:10.3390/s23073565_

Round 1

Reviewer 1 Report

Authors have compared existing machine learning algorithms to detect stress using bracelets.

1.       Line 12: what are the two accuracies 73.3 and 75.9?

2.       Line 65: What is the meaning of transfer disease and why stress is a transfer disease?

3.       Line 92: Abbreviation should be explained whenever first used.

4.       In the introduction section, the authors may describe the novelty of the author’s work.

5.       Figure 1: how can we make sure that this protocol will put stress on a person? Stress is different from tiredness. Is there any reference from medical science to confirm that this protocol induces stress?

6.       Section 1.2: In the related work a comparative table should highlight the main contributions of the published literature. Moreover different stress-inducing protocols should be highlighted.

7.       Table 1: it is better to include the equations or methods to obtain the features from the raw data.

8.       Section 2.5 is not a classification problem. It describes the performance metrics.

9.       Classification algorithms used in the study should also be described. Also, show how the hyperparameters of these methods are tuned.

10.   For ten-fold cross-validations, all the results should be reported as mean and standard deviation (Tables 2 and 3).

11.   What is the effect of the feature reduction method?

12.   Performance of these algorithms must be compared with the published results.

Reviewer 2 Report

Dear authors,

Thank you for submitting your paper titled "A Method for Stress Detection using Empatica E4 bracelet and Machine Learning Techniques" for review. I have read your paper and have the following comments:

Positives:

  • The study is well-designed and the methodology is clear and well-explained.
  • The paper provides useful insights into the selection of features for characterizing stress, the performance of various machine learning algorithms, and the importance of addressing the issue of an unbalanced dataset.
  • The paper is well-written and structured.
  • The figures and tables provide a clear and concise summary of the results.
  • The choice of machine learning algorithms and feature selection methods is appropriate.

Areas of improvement:

  • The introduction could be more clearly written to provide a background and context for the study.
  • The section on feature extraction could be expanded to provide more details on the specific methods used and how they were chosen.
  • The Results section is short and lacks a discussion about the trends and other findings in the figures. Please provide more discussion in this section.
  • The study has some limitations, including a small sample size and the use of a laboratory environment. Please discuss these limitations in more detail in the paper.
  • The figures could be improved by using larger and clearer fonts.
  • it would be valuable to include figures showing the training accuracy and loss vs validation to demonstrate the model's immunity to overfitting, especially given the small number of samples in the dataset. Additionally, confusion matrices for the test dataset would provide further insights into the performance of the model.

Overall, your paper provides valuable insights into the use of physiological signals to assess mental stress using wearable sensors. With some minor revisions and additions, I believe that your paper could make a valuable contribution to the field of stress monitoring. Thank you for your contribution, and I look forward to seeing your future work.

Best regards,

Reviewer 3 Report

The paper is clear and the topic is interesting and current. The paper mentions that the considered classification algorithms have been selected based on the state of the art presented in section 1.2. It would however be interesting to have some additional details on why these specific classification algorithms, and not others, have been selected.

Round 2

Reviewer 1 Report

The authors have revised the paper and answered some of my comments. But still, there are a few comments that should be addressed in the revised version,

1.       Weka and Matlab learner are two libraries of machine learning algorithms. What is the purpose of applying these two libraries to apply the same algorithm on the dataset? Authors should report all the analyses based on one library only if the same algorithms are used.

2.       “We have described in the “Data acquisition protocol” subsection all the different categories of stressors found in literature to strength the validity of our protocol”. So what type of stress you are using and does the protocol is supported by any evidence that it will produce the stress. Moreover, how you have combined all the stresses?

3.       Line 279-280, “To extract this information, an algorithm was developed using several functions available on Matlab Statistics and Machine Learning Toolbox.” This statement is not enough. It is important to explain the feature extraction equations or methods from the data. Show mathematical equations to calculate these features.

4.       Hyperparameters are very important for the performance of a classification algorithm. Therefore, it is important to give the values of the important hyperparameters of all the classification algorithms.

5.       Lines 297-305, the Explanation of the feature extraction method is not very clear. Please describe the by a pseudocode or algorithm or set of mathematical equations. Why two different selection methods are used and what is their effect on the performance of the classification? So provides tables to show performance metrics for a full set of features and a reduced set of features.

6.       It is also important to compare your results with the published results to show the efficacy of your proposed framework.

Round 3

Reviewer 1 Report

The authors have addressed all my comments in the revised version. Now the revised version is acceptable.